# Small Bowel Metastatic Melanoma: An Emblematic “Coal-Black” Appearance at Videocapsule Endoscopy

**DOI:** 10.3390/medicina57121313

**Published:** 2021-11-30

**Authors:** Alessia Todeschini, Ilaria Loconte, Antonella Contaldo, Enzo Ierardi, Alfredo Di Leo, Mariabeatrice Principi

**Affiliations:** Section of Gastroenterology, Department of Emergency and Organ Transplantation, University “Aldo Moro” of Bari, 70124 Bari, Italy; ilarialoconte@libero.it (I.L.); contaldoantonella@gmail.com (A.C.); ierardi.enzo@gmail.com (E.I.); alfredo.dileo@uniba.it (A.D.L.); b.principi@gmail.com (M.P.)

**Keywords:** small bowel, capsule endoscopy, melanoma

## Abstract

A 80-year-old woman underwent vulvar melanoma resection and segmental lung resection for pulmonary metastasis. Immunotherapy with Nivolumab was performed. One year later, the patient was admitted for gastrointestinal (GI) recurrent bleeding and severe anemia. Esophagoastroduodenoscopy and colonoscopy did not show any abnormality, while videocapsule endoscopy (VCE) revealed an irregular and exophytic whitish area with a “coal-black” central depression. Small bowel resection was performed and histological examination revealed S100 protein strongly positive melanoma metastasis. The patient died six months later from disease progression. A “coal-black” appearance of intestinal metastatic melanoma has been described only twice before this report. In one case the patient had been treated by immunotherapy with interferon A and dendritic cell-based vaccination. In our patient, it is presumable that the picture we observed was a consequence of Nivolumab treatment inducing the disappearance of melanocytes in the area surrounding the metastasis with the onset of the central coal-black lesion encircled by whitish tissue. This picture should be emblematic of intestinal metastatic melanoma in subjects treated with immunotherapy showing occult/obscure bleeding.

## 1. Introduction

VCE is a non-invasive device, which allows investigating small bowel lesions, especially in patients with unexplained iron deficiency anemia and obscure/occult bleeding [1]. Among potential sources of intestinal hemorrhage, cutaneous melanoma metastasis deserves a special mention. Indeed, despite the GI tract being the most common site of metastastic localization of this tumor, symptomatic involvement is found only from 0.8% to 4.7%, since its finding occurs postmortem in more than 60% of cases [2]. GI bleeding, perforation and intestinal obstruction are the most frequent manifestations and VCE constitutes the most suitable diagnostic tool [3]. GI metastatic melanomas are classified as submucosa-like or primary carcinoma-like tumors and lesions may appear as ulcerated with or without signs of bleeding [4]. Smedegaard and Adamsen in 2007 firstly described a singular VCE picture of intestinal metastatic melanoma, i.e., a characteristic “coal-black” appearance, which had never been previously described in intestinal tumors [5]. Subsequently, in 2010, Urbain et al. showed a similar feature and marked out the coal-black lesion as having a “solar eclipse” appearance, highlighting that this picture is typically representative of melanoma metastasis undergoing immunotherapy [6].

Hereby, we aim to report the imaging series of the third case of a “coal-black” appearance of intestinal metastatic melanoma.

## 2. Case Report

An 80-year-old woman was submitted to vulvar melanoma resection and bilateral inguinal lymphadenectomy in 2016. The diagnosis was obtained by surgical specimen histological examination. Since the operation time, a strict follow-up was started by computed tomography (CT) scan of the brain, thorax and abdomen every six months and a total positron emission tomography (PET) scan once a year. In 2018, a timely screening chest CT scan showed a pulmonary micro-nodule in the left inferior lobe. The patient underwent segmental lung resection and histology confirmed pulmonary metastasis of melanoma with pleural infiltration. Subsequently, immunotherapy with anti-PD1 agent Nivolumab was started. One year later, the patient was admitted to Oncology Unit of University Hospital Policlinico, Bari, Italy for GI recurrent bleeding and severe anemia needing blood transfusion. Esophagoastroduodenoscopy and colonoscopy were performed at Gastroenterology Unit and bone marrow biopsy at Hematology Unit in the same hospital; none of these investigations showed any abnormality. Therefore, VCE (Pillcam SB-Given, Medtronic, Dublin, Ireland) was performed (Gastroenterology Unit, University Hospital Policlinico, Bari, Italy). Two hours and fourteen minutes after ingestion, imaging revealed an irregular and exophytic whitish area with a “coal-black” central depression strongly indicative of melanoma metastasis (Figure 1 and Appendix A). Therefore, small bowel resection was performed and histological examination confirmed the diagnosis of melanoma. Additionally, immunohistochemistry revealed S100 protein strongly positive melanoma, thus confirming the metastatic origin of the lesion (Figure 2), since this immunohistochemical marker is specific in detecting metastasis of melanoma [7]. Postoperative course was good and the patient was discharged within a few days. She was scheduled to continue immunotherapy at Oncology Unit of University Hospital Policlinico, Bari, Italy. Despite this diagnostic and therapeutic approach, the patient died six months later from disease progression.

## 3. Discussion

A study enclosing 5129 VCE investigations showed that 66% of secondary small bowel tumors are melanomas [8]. Nevertheless, most small bowel melanoma metastases are asymptomatic and represent a postmortem finding [2]. Presumably, they are underdiagnosed, since the examination of the small bowel is not included in the current guidelines and is performed, as in our patient, only in the case of complications. Therefore, it would be worth considering whether a timely screening with VCE at the time of diagnosis of melanoma could improve the survival of patients. Indeed, our patient had a rapid disease progression and low survival, presumably for a late intestinal accurate investigation. Consequently, it may be of relevance not only to investigate small bowel with VCE in the follow up of melanoma, but also to improve the information about the endoscopic appearance of this neoplasm, especially under immunotherapy. The “coal-black” appearance at VCE has been described only twice to the best of our knowledge and this aspect has been reported as emblematic. On these bases, our demonstration of a further case may be significant in order to highlight a diagnostic approach to intestinal metastatic melanoma, since clinicians should be aware of this rare, but potential endoscopic picture for two main reasons. First, it is typically representative of this pathological condition. Additionally, it may be expected in subjects undergoing immunotherapy [6]. The possibility of metastatic intestinal melanoma occurrence even after specific immunotherapy has been emphasized by Miyazawa et al., who described two cases of melanoma intestinal metastasis despite other metastatic sites having been completely responsive to Nivolumab and despite the patients having favorable findings, such as vitiligo and normal lactate dehydrogenase [9]. In our case, similarly, metastatic intestinal localization was found one year from when immunotherapy with anti-PD1 agent Nivolumab was started.

## 4. Conclusions

As previously hypothesized in the case of Urbain et al. treated with interferon A and dendritic cell-based vaccination [6], it is also possible that in our patient, the unusual, even if emblematic, “solar eclipse” appearance was a consequence of Nivolumab treatment. Presumably, this agent induced the disappearance of melanocytes in the area surrounding the metastasis with the onset of whitish intestinal tissue surrounding a central coal-black lesion. On these bases, this picture should be accurately detected in subjects with a history of melanoma and immunotherapy showing iron deficiency anemia due to occult/obscure bleeding. Additionally, this picture could be found even in patients with melanoma under immunotherapy, regardless of the occurrence of intestinal bleeding, perforation and obstruction.

## Figures and Tables

**Figure 1 medicina-57-01313-f001:**
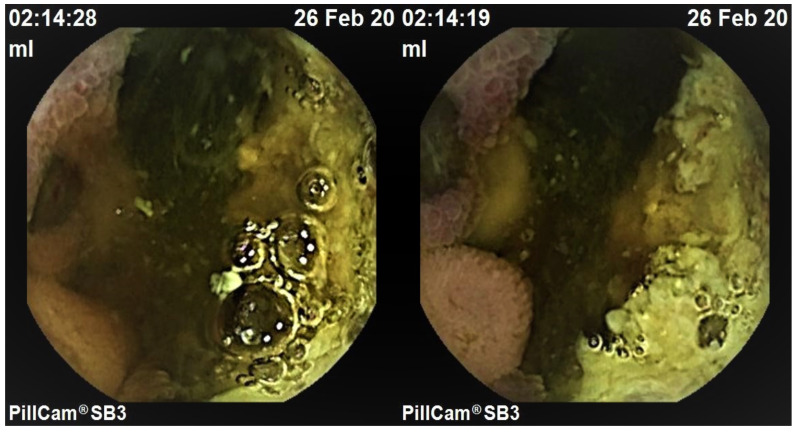
VCE appearance of characteristic coal-black central depression surrounded by exophytic whitish area emblematic of metastatic melanoma in the small bowel.

**Figure 2 medicina-57-01313-f002:**
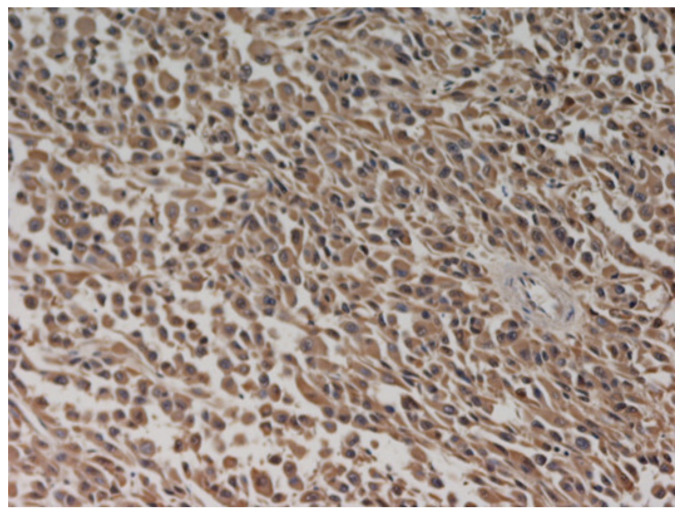
Immunohistochemical picture of S100 protein strongly positive melanoma metastasis (hematoxylin counterstain; 400×).

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
