# Peer review of "Small Bowel Metastatic Melanoma: An Emblematic “Coal-Black” Appearance at Videocapsule Endoscopy"

_medicina, 2021, doi:10.3390/medicina57121313_

Round 1

Reviewer 1 Report

Overall, this is an interesting case report which describe about intestinal metastatic melanoma through VCE. I only have several minor comments regarding this case report:

  1. At the end of the Introduction section, the authors should give some statements regarding the purpose/aim of this report to increase the significance of this paper. For example, "Hereby, in this study, we aim to report the third case of "coal-black" appearance of intestinal metastatic melanoma."
  2. The settings (name of the hospitals) for this case should be mentioned.
  3. The historical timeline of the patients should be described more. Do the patients underwent laboratory or radiological examination in 2016 before the resection of vulvar melanoma? In 2018, what made the physician performed a chest CT-scan for the patient? Considering that GI tract is one of the common organ for melanoma metastatic, did any abdominal examination (endoscopy, CT-scan/MRI, PET scan) performed in 2018? etc
  4. The final treatment for the patients after bowel resection should also be described and discussed in the Discussion section.

Author Response

  1. In revised manuscript, a sentence was enclosed according to reviewer’s suggestion.
  2. In revised manuscript, requested details have been reported.

  3. In detail, the diagnosis was obtained by surgical specimen histological examination. Since operation time, a strict follow-up was started by computed tomography (CT) scan of the brain, thorax and abdomen every six months and total positron emission tomography (PET) scan once a year. In 2018, timely screening chest CT scan showed a pulmonary micro-nodule in the left inferior lobe. 

    Of note, we emphasize that the patient was handled at our Gastroenterology Unit solely for the endoscopic investigations (esophagogastroduodenoscopy, colonoscopy and video capsule).

  4. Postoperative course was good and the patient was discharged within few days. She was scheduled to continue immunotherapy at Oncology Unit of University Hospital Policlinico, Bari, Italy. Despite this diagnostic and therapeutic approach, the patient died six months later for disease progression. On the bases of this experience, we discussed in revised manuscript whether VCE could be enclosed in follow up investigations of patients with melanoma in order to obtain an early diagnosis of intestinal metastasis and improve survival.

    Again,  we emphasize that the patient was handled at our Unit solely for the endoscopic investigations (esophagogastroduodenoscopy, colonoscopy and video capsule).

Reviewer 2 Report

I truly enjoyed reading the case report by Todeschini and colleagues and I think it is suitable for publication in its current form.

In summary, they describe a case of small bowel melanoma, which was diagnosed by video capsule endoscopy (VCE) indicated for recurrent GI bleeding which occurred two years after resection of the primary and lung metastasis, and one year after immunotherapy.

Immunotherapy is a treatment modality, which is increasingly used in melanoma and other tumors, thus we will most likely see more of these cases in the future. As the authors state correctly, small bowel is a common site for melanoma metastases and is certainly underdiagnosed. VCE, on the other hand, is not only a suitable technique for obscure GI bleeding, but is under investigation for many other indications. Hence, there is still much to learn for image interpretation.

In this sense, the current article is important (1) to further increase the awareness of melanoma metastases in the small bowel and, (2) to spread knowledge of how these metastases could look like, especially under immunotherapy.

In addition, the authors discuss that SB metastases might appear even if other metastatic sites respond adequately to immunotherapy – an intriguing fact that could motivate researchers to build new hypothesis on the particular immune microenvironment of the intestine.

There is only point that I missed in the discussion:

It seems true that most SB metastases remain asymptomatic, which is why they are underdiagnosed, and yet examination of the SB is not included in the current guidelines. Do the authors believe that a timely screening with VCE at the time of diagnosis (and before the first GI bleeding occurred) would have improved the survival of their patient? In other words, should VCE part of the diagnostic work-up of (metastatic) melanoma patients in their opinion? I suggest that the authors consider commenting on this point in the discussion.

Finally, I have not detected any formal errors in the manuscript. I hereby declare, that I have no conflicts of interest with publishing this paper.   

Author Response

Thank you very much for the kind appreciation of our report. As suggested, we discussed whether a timely screening with VCE at the time of diagnosis of melanoma could improve the survival of patients by an early detection of intestinal metastasis in revised manuscript.